# QTN-VQC: An End-to-End Learning framework for Quantum Neural Networks

## Abstract

The advent of noisy intermediate-scale quantum (NISQ) computers raises a crucial challenge to design quantum neural networks for fully quantum learning tasks. To bridge the gap, this work proposes an end-to-end learning framework named QTN-VQC, by introducing a trainable quantum tensor network (QTN) for quantum embedding on a variational quantum circuit (VQC). The architecture of QTN is composed of a parametric tensor-train network for feature extraction and a tensor product encoding for quantum embedding. We highlight the QTN for quantum embedding in terms of two perspectives: (1) we theoretically characterize QTN by analyzing its representation power of input features; (2) QTN enables an end-to-end parametric model pipeline, namely QTN-VQC, from the generation of quantum embedding to the output measurement. Our experiments on the MNIST dataset demonstrate the advantages of QTN for quantum embedding over other quantum embedding approaches.

## 1 Introduction

The state-of-the-art machine learning (ML), particularly based on deep neural networks (DNN), has enabled a wide spectrum of successful applications ranging from the everyday deployment of speech recognition (Deng et al., 2013) and computer vision (Sermanet et al., 2014) through to the frontier of scientific research in synthetic biology (Jumper et al., 2021). Despite rapid theoretical and empirical progress in DNN based regression and classification (Goodfellow et al., 2016), DNN training algorithms are computationally expensive for many new scientific applications, such as new drug discovery (Smalley, 2017), which requires computational resources that are beyond the computational limits of classical hardwares (Freedman, 2019). Fortunately, the imminent advent of quantum computing devices opens up new possibilities of exploiting quantum machine learning (QML) (Biamonte et al., 2017; Schuld et al., 2015; Schuld & Petruccione, 2018; Schuld & Killoran, 2019; Saggio et al., 2021; Dunjko, 2021) to improve the computational efficiency of ML algorithms in the new scientific domains.

Although the exploitation of quantum computing devices to carry out QML is still in its initial exploratory stages, the rapid development in quantum hardware has motivated advances in quantum neural networks (QNN) to run in noisy intermediate-scale quantum (NISQ) devices (Preskill, 2018; Huggins et al., 2019; Huang et al., 2021; Kandala et al., 2017). A NISQ device means that not enough qubits could be spared for quantum error correction, and the imperfect qubits have to be directly used at the physical layer. Even though, a compromised QNN approach is proposed by employing hybrid quantum-classical models that rely on the optimization of variational quantum circuits (VQC) (Benedetti et al., 2019; Mitarai et al., 2018). The resilience of the VQC based models to certain types of quantum noise errors and high flexibility concerning coherence time and gate requirements (McClean et al., 2018) admit many practical implementations of QNN on NISQ devices (Chen et al., 2020b; Yang et al., 2021; Du et al., 2020; 2021; Skolik et al., 2021; Dunjko et al., 2016; Jerbi et al., 2021; Ostaszewski et al., 2021). One notable limitation in the current QNN training pipeline is that the quantum embedding is not fully realizable in a quantum computer, which may impede the learning of the QNN. Hence, this work proposes QTN-VQC to enable an end-to-end trainable QNN, including data embedding to quantum measurements, that are easily realizable in quantum devices, where QTN stands for the quantum tensor network (Orús, 2019; Huckle et al., 2013; Biamonte et al., 2017; Murg et al., 2010) for generating quantum embedding.

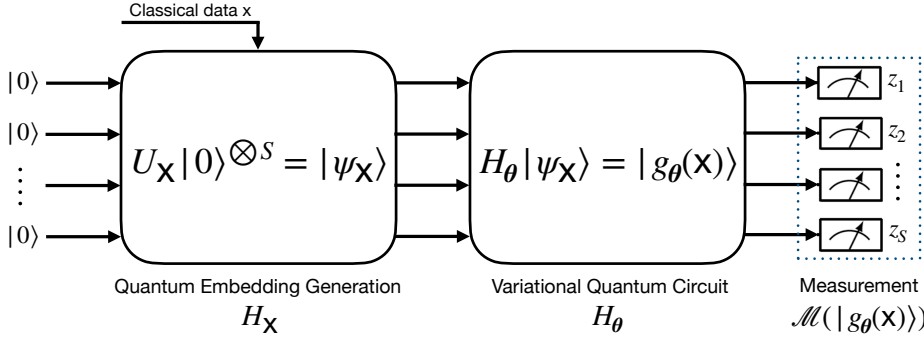

Figure 1: *An illustration of QNN based on VQC.*

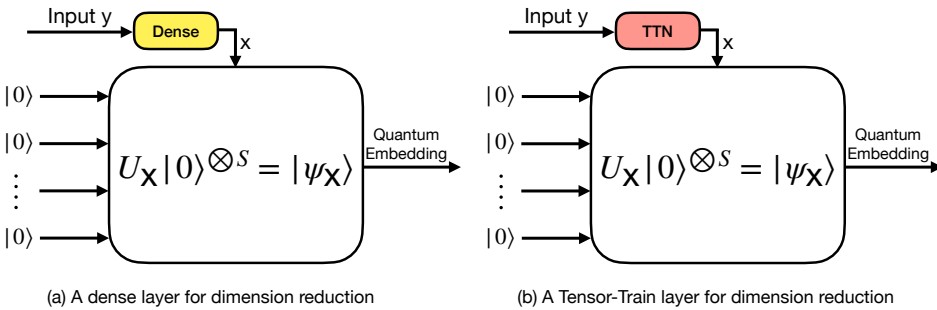

(a) A dense layer for dimension reduction    (b) A Tensor-Train layer for dimension reduction

Figure 2: *Different paradigms for quantum embedding. (a) a dense layer is used to generate low-dimensional vector $\mathbf{x}$ from a high-dimensional one $\mathbf{y}$; (b) a TTN is used for dimension reduction.*

As shown in Figure 1, our QNN builds a unitary linear operator that consists of three main components: (1) quantum embedding generation; (2) variational quantum circuit; (3) measurement. Quantum embedding generation, also known as quantum encoding, applies a fixed unitary linear operator $H_{\mathbf{x}}$ transforming classical vectors $\mathbf{x}$ to quantum states $|\psi_{\mathbf{x}}\rangle$ in a Hilbert space. This step is an important aspect of designing quantum algorithms that directly impact the entire computation cost of VQC and owns a characteristic of quantum superposition. Moreover, the VQC comprises two types of quantum gates: (1) Controlled-NOT (CNOT) gates; (2) learnable parametric quantum gates. The CNOT gates ensure the property of quantum entanglement through mutually connecting the qubits, and the parametric quantum gates can be adjustable to best fit the quantum input states. The model parameters of VQC should be optimized by employing variants of gradient descent algorithms during the training process. Those parametric quantum gates of VQC are similar to the weights assigned to DNN, and such quantum circuits have been justified to be resilient to quantum noises (Farhi et al., 2014; Kandala et al., 2017; McClean et al., 2016). Besides, the measurement $\mathcal{M}(|g_{\boldsymbol{\theta}}(\mathbf{x})\rangle)$ aims at projecting the quantum output states $|g_{\boldsymbol{\theta}}(\mathbf{x})\rangle$ to one classical output $z_i$.

This work focuses on quantum embedding generation because it is quite related to the practical usage in machine learning applications in terms of computational cost and representation capability of classical input features. In particular, we design a novel quantum tensor network (QTN) for quantum embedding generation. More specifically, the QTN consists of a tensor-train network (TTN) for dimension reduction and a quantum tensor encoding framework for outputting quantum embeddings. The dimension reduction is a necessary procedure before the quantum encoding because only a small number of qubits could be supported on available NISQ computers at this moment. A typical approach for dimension reduction relies on a classical fully-connected layer, also known as a dense layer, to convert high-dimensional input vectors $\mathbf{y}$ into low-dimensional ones $\mathbf{x}$. However, since a dense layer cannot be physically mapped on a quantum computer, much overhead has to be incurred by frequently communicating between classical and quantum devices during the end-to-end training pipeline.

As shown in Figure 2 (b), one of our **contribution** is to leverage a tensor train network (TTN) to replace the dense layer in Figure 2 (a). The benefits of applying TTN arise from two aspects: (1) TTN can maintain the representation power of the dense layer, which will be justified in our theorems; (2) TTN is a tensor network and can be flexibly placed in quantum computers, which enables an end-to-end training process fully conducted in a quantum computer. Moreover, in this work, a tensor product encoding (TPE) is delicately designed for generating quantum embedding, which builds the relationship between a classical vector $\mathbf{x}$ and the corresponding quantum state $|\mathbf{x}\rangle$; Besides, we further investigate the representation of QTN-VQC in terms of model size and non-linear activation function used in TTN. We denote a QTN as the combination of TTN and TPE and utilize QTN-VQC as a genuine end-to-end learning framework for QNN.

## 2 RELATED WORK

The work (Schuld & Petruccione, 2018; Biamonte et al., 2017; Dunjko & Briegel, 2018) demonstrate that VQC shows great promise in surpassing the performance of classical ML. Prominent examples of VQC based models include quantum approximate optimization algorithm (QAOA) (Farhi et al., 2014), and quantum circuit learning (QCL) (Mitarai et al., 2018). Various architectures and geometries of VQC have been shown in tasks ranging from image classification (Henderson et al., 2020; Chen et al., 2020a; Kerenidis et al., 2020) to reinforcement learning (Chen et al., 2020b).

As for quantum embedding, basis encoding is the process of associating classical input data in the form of binary strings with the computational basis state of a quantum system (Leymann & Barzen, 2020). Similarly, amplitude encoding is a technique involving encoding data into the amplitudes of a quantum state (Soklakov & Schack, 2006). Unfortunately, the computational cost of both quantum embedding and amplitude encoding becomes exponentially expensive with the increasing number of qubits (Schuld & Killoran, 2019). A new technique of angle embedding makes use of the quantum gates to generate quantum states (Fu et al., 2011), but it cannot deal with the high-dimensional feature inputs. Therefore, this work exploits the use of TTN for dimension reduction followed by a TPE for generating quantum embedding.

In particular, this work employs the TTN for dimensionality reduction. The TTN model based on TT decomposition in neural networks was first proposed in (Oseledets, 2011), and it could be flexibly extended the convolutional neural network (CNN) (Garipov et al., 2016) and recurrent neural network (RNN) (Tjandra et al., 2017). The empirical study of TTN on machine learning tasks shows that TTN is capable of maintaining the DNN baseline results (Qi et al., 2020a; Yu et al., 2017; Yang et al., 2017; Jin et al., 2020). However, to our best knowledge, no existing works have applied TTN to QML. Besides, since the tensor network-based machine learning model like TTN is closely related to quantum machine learning in terms of their model structures (Liu & Wang, 2018; Gao et al., 2017), the QTN-VQC model can be directly regarded as the classical simulation of the corresponding quantum machine learning. In addition to a classical dense layer, more complicated architectures like AlexNet (Lloyd et al., 2020) could be used for dimension reduction, and we also compare the performance between TTN and AlexNet-based models.

## 3 NOTATIONS

We denote $\mathbb{R}^I$ as a $I$-dimensional real coordinate space, and $\mathbb{R}^{I_1 \times I_2 \times \cdots I_K}$ refers to a space of $K$-order tensors. The symbol $\mathcal{W} \in \mathbb{R}^{I_1 \times I_2 \times \cdots \times I_K}$ represents a $K$-order multi-dimensional tensor in $\mathbb{R}^{I_1 \times I_2 \times \cdots I_K}$, and the symbols $\mathbf{v} \in \mathbb{R}^I$ and $\mathbf{W} \in \mathbb{R}^{I \times J}$ represent a vector and a matrix, respectively.

For the notations of quantum computing, $\forall \, \mathbf{v} \in \mathbb{R}^I$, the symbol $|\mathbf{v}\rangle$ denotes a quantum state associated with a $2^I$-dimensional vector in a Hilbert space. Particularly, $|0\rangle = [1 \ 0]^T$ and $|1\rangle = [0 \ 1]^T$.

The quantum gate $R_Y(\theta)$ means a Pauli-$Y$ gate with a unitary operator as defined in Eq. (1), which implies a qubit rotates the Bloch sphere along the Y-axis by a given angle $\theta$.

$$R_Y(\theta) = \begin{bmatrix} \cos\frac{\theta}{2} & -i\sin\frac{\theta}{2} \\ -i\sin\frac{\theta}{2} & \cos\frac{\theta}{2} \end{bmatrix} \tag{1}$$

Moreover, the operator $\otimes$ is a tensor product. Given the vectors $\mathbf{v}_i \in \mathbb{R}^I$, the tensor product of $I$ vectors is defined as $\otimes_{i=1}^I \mathbf{v}_i$, which is a $2^I$-dimensional vector and can provide a compact represen-

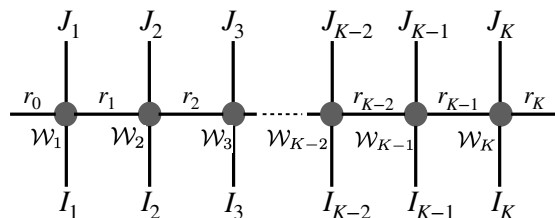 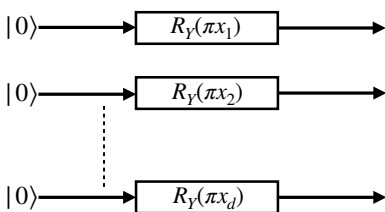

(a) Tensor-Train Network: given a set of TT-ranks $\{r_1, r_2, \ldots, r_K\}$, a circle represents a core tensor and each line is related to a dimension.

(b) Tensor Product Encoding: $R_Y(\pi x)$ denotes the Pauli-Y rotation gate

Figure 3: *A demonstration of quantum tensor network for quantum embedding.*

tation for $\mathbf{v}_1 \otimes \mathbf{v}_2 \otimes \cdots \otimes \mathbf{v}_I$. Similarly, the symbol $|0\rangle^{\otimes S}$ means a tensor product of $S$ quantum states of $|0\rangle$. Furthermore, for a scalar $v$, the quantum state $|v\rangle$ can be written as:

$$|v\rangle = \cos v |0\rangle + \sin v |1\rangle = \begin{bmatrix} \cos v \\ \sin v \end{bmatrix}. \tag{2}$$

# 4 QTN-VQC: OUR PROPOSED END-TO-END LEARNING FRAMEWORK

This section introduces our proposed end-to-end learning framework, namely QTN-VQC in this work. As shown in Figure 3, the QTN model includes two components (a) TTN and (b) TPE, which will be separately introduced in Section 4.1 and Section 4.2. Moreover, Figure 4 illustrates the framework of VQC and Section 4.3 is devoted to discussing the details of VQC.

## 4.1 TENSOR TRAIN NETWORK FOR DIMENSION REDUCTION

We leverage TTN (Novikov et al., 2015) for the dimension reduction of input features. TTN relies on the TT decomposition (Oseledets, 2011) and has been commonly employed in machine learning tasks like speech processing (Qi et al., 2020b) and computer vision (Yang et al., 2017). The TT decomposition assumes that given a set of TT-ranks $\{r_0, r_1, ..., r_K\}$, a $K$-order tensor $\mathcal{W} \in \mathbb{R}^{I_1 \times I_2 \times \cdots \times I_K}$ is factorized into the multiplication of 3-order tensors $\mathcal{X}_k \in \mathbb{R}^{r_{k-1} \times I_k \times r_k}$. In more detail, given a set of indices $\{i_1, i_2, ..., i_K\}$, $\mathcal{X}(i_1, i_2, ..., i_K)$ is decomposed as:

$$\mathcal{X}(i_1, i_2, ..., i_K) = \prod_{k=1}^{K} \mathcal{X}_k(i_k), \tag{3}$$

where $\forall i_k \in [I_k]$, $\mathcal{X}_k(i_k) \in \mathbb{R}^{r_{k-1} \times r_k}$. Since $r_0 = r_1 = 1$, the term $\prod_{k=1}^{K} \mathcal{X}_k(i_k)$ is a scalar value.

TTN employs the TT decomposition in a dense layer and is explicitly demonstrated in Figure 3 (a). In more detail, for an input tensor $\mathcal{X} \in \mathbb{R}^{I_1 \times I_2 \times \cdots \times I_K}$ and an output tensor $\mathcal{Y} \in \mathbb{R}^{J_1 \times J_2 \times \cdots \times J_K}$, we achieve

$$
\begin{aligned}
\mathcal{Y}(j_1, j_2, ..., j_K) &= \sum_{i_1=1}^{I_1} \sum_{i_2=1}^{I_2} \cdots \sum_{i_K=1}^{I_K} \mathcal{W}\left((i_1, j_1), (i_2, j_2), ..., (i_K, j_K)\right) \mathcal{X}(i_1, i_2, ..., i_K) \\
&= \sum_{i_1=1}^{I_1} \sum_{i_2=1}^{I_2} \cdots \sum_{i_K=1}^{I_K} \left(\prod_{k=1}^{K} \mathcal{W}_k(i_k, j_k)\right) \cdot \prod_{k=1}^{K} \mathcal{X}_k(i_k) \\
&= \prod_{k=1}^{K} \left(\sum_{i_k=1}^{I_k} \mathcal{W}_k(i_k, j_k) \mathcal{X}_k(i_k)\right) \\
&= \prod_{k=1}^{K} \mathcal{Y}_k(j_k),
\end{aligned}
\tag{4}
$$

where $\mathcal{X}_k(i_k) \in \mathbb{R}^{r_{k-1} \times r_k}$, and $\mathcal{Y}_k(j_k) \in \mathbb{R}^{r_{k-1} \times r_k}$ which results in a scalar $\prod_{k=1}^{K} \mathcal{Y}_k(j_k)$ because of the ranks $r_0 = r_1 = 1$; $\mathcal{W}((i_1, j_1), (i_2, j_2), ..., (i_K, j_K))$ is closely associated with $\mathcal{W}(m_1, m_2, ..., m_K)$ as defined in Eq. (3), if each index $m_k = i_k \times j_k$ is set. The multi-dimensional tensor $\mathcal{W}$ is decomposed into the multiplication of 4-order tensors $\mathcal{W}_k \in \mathbb{R}^{r_{k-1} \times I_k \times J_k \times r_k}$. A nonlinear activation function, e.g., Sigmoid, Tanh, and ReLU, is imposed upon the tensor $\mathcal{Y}$. Compared with a dense layer with $\prod_{k=1}^{K} I_k J_k$ parameters, a TTN owns as few as $\sum_{k=1}^{K} r_k r_{k-1} I_k J_k$ trainable parameters.

When a TTN is utilized for the dimension reduction, the high-dimensional input vector $\mathbf{x} \in \mathbb{R}^I$ is first reshaped into a tensor $\mathcal{X} \in \mathbb{R}^{I_1 \times I_2 \times \cdots \times I_K}$, and then we can represent $\mathcal{X}$ as a TT format that goes through TTN. The outputs of TTN can be converted back to a tensor $\mathcal{Y} \in \mathbb{R}^{J_1 \times J_2 \times \cdots \times J_K}$, which is further reshaped to a lower dimensional vector $\mathbf{y} \in \mathbb{R}^J$. Here, we define $\prod_{k=1}^{K} I_k = I$ and $\prod_{k=1}^{K} J_k = J$. Moreover, the computational complexities of TTN and the related dense layer are in the same scale, which is discussed in (Yang et al., 2017).

Eq. (4) suggests that TTN is a multi-dimensional extension of a dense layer, where the trainable weight matrix of a dense layer is changed to the learnable core tensors. Additionally, many empirical studies demonstrate that a TTN is capable of maintaining the baseline results of the dense layer (Qi et al., 2020b; Yang et al., 2017; Novikov et al., 2015; Qi et al., 2020a). More significantly, since TTN can be flexibly mapped into a quantum circuit, the quantumness inherent in TTN brings great advantages over other architectures like the dense layer. In other words, although TTN is treated classically, it is possible to substitute equivalent quantum circuits for TTN when more qubits become available (Du et al., 2020), which implies that QTN-VQC stands for a genuine end-to-end QNN learning architecture on a quantum computer.

Furthermore, since the gradient exploding and diminishing problems are serious issues in the TTN training. To avoid those training problems, we only consider 3-order core tensors and small TT-ranks to configure a simple TTN in our experimental simulations. Our theoretical analysis of QTN-VQC based on Theorem 3 in Section 5 suggests that the representation power is not related to TT-ranks and the tensor order $K$, thus small TT-ranks and the tensor order $K$ are preferred. In particular, a lower $K$ can significantly reduce the computational cost and speed up the convergence rate.

## 4.2 TENSOR PRODUCT ENCODING

In this subsection, we first introduce Theorem 1, and then we derive our TPE associated with the circuits in Figure 3 (b).

**Theorem 1.** *Given the classical vector $\boldsymbol{x} = [x_1, x_2, ..., x_I]^T \in \mathbb{R}^I$, a TPE as shown in Figure 3 (b) can result in a quantum state $|\boldsymbol{x}\rangle$ with the following complete vector representation as:*

$$\left(\otimes_{i=1}^{I} R_Y(2x_i)\right) |0\rangle^{\otimes I} = \begin{bmatrix} \cos x_1 \\ \sin x_1 \end{bmatrix} \otimes \begin{bmatrix} \cos x_2 \\ \sin x_2 \end{bmatrix} \otimes \cdots \otimes \begin{bmatrix} \cos x_I \\ \sin x_I \end{bmatrix} = |\boldsymbol{x}\rangle. \tag{5}$$

*Proof.* Since each element $x_i$ in the vector $\mathbf{x}$ can be written as $|x_i\rangle = \cos x_i |0\rangle + \sin x_i |1\rangle$, the quantum state $|\mathbf{x}\rangle$ can be written as:

$$|\mathbf{x}\rangle = \begin{bmatrix} \cos x_1 \\ \sin x_1 \end{bmatrix} \otimes \begin{bmatrix} \cos x_2 \\ \sin x_2 \end{bmatrix} \otimes \cdots \otimes \begin{bmatrix} \cos x_I \\ \sin x_I \end{bmatrix}. \tag{6}$$

When the vector $\mathbf{x}$ goes through the quantum tensor network, which implies the following as:

$$R_Y(2x_i)|0\rangle = cos x_i|0\rangle + sin x_i|1\rangle = |x_i\rangle. \tag{7}$$

The preceding equation, in turn, implies that Eq. (5). □

Theorem 1 builds a connection between the vector $\mathbf{x}$ and the quantum state $|\mathbf{x}\rangle$, and the resulting $|\mathbf{x}\rangle$ is taken as the quantum embedding as the inputs to VQC. Since $\otimes_{i=1}^{I} R_Y(2x_i)$ is a reversely unitary linear operator, there is no information loss incurred during the stage of quantum encoding. Furthermore, if the input is multiplied with a constant $\frac{\pi}{2}$, we obtain the following term as:

$$\left(\otimes_{i=1}^{I} R_Y(\pi x_i)\right) |0\rangle^{\otimes I} = \begin{bmatrix} \cos(\pi x_1) \\ \sin(\pi x_1) \end{bmatrix} \otimes \begin{bmatrix} \cos(\pi x_2) \\ \sin(\pi x_2) \end{bmatrix} \otimes \cdots \otimes \begin{bmatrix} \cos(\pi x_I) \\ \sin(\pi x_I) \end{bmatrix}, \tag{8}$$

which corresponds to Figure 3 (b).

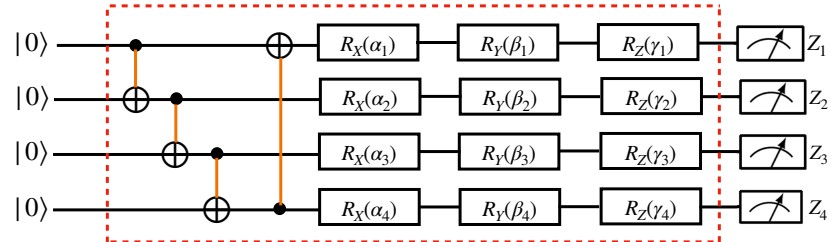

(a) Variational quantum circuit: the dashed square indicates repeated model with CNOT gates for entangling quantum states, and $R_X(\alpha)$, $R_Y(\beta)$, $R_Z(\gamma)$ represent Pauli-X, Y, Z gates with free parameters $\alpha, \beta, \gamma$

$$R_Y(\beta) = \begin{bmatrix} \cos\frac{\beta}{2} & -\sin\frac{\beta}{2} \\ \sin\frac{\beta}{2} & \cos\frac{\beta}{2} \end{bmatrix} \qquad \begin{bmatrix} 1 & 0 & 0 & 0 \\ 0 & 1 & 0 & 0 \\ 0 & 0 & 0 & 1 \\ 0 & 0 & 1 & 0 \end{bmatrix} \qquad R_X(\alpha) = \begin{bmatrix} \cos\frac{\alpha}{2} & -i\sin\frac{\alpha}{2} \\ -i\sin\frac{\alpha}{2} & \cos\frac{\alpha}{2} \end{bmatrix} \qquad R_Z(\gamma) = \begin{bmatrix} \exp(-i\frac{\gamma}{2}) & 0 \\ 0 & \exp(i\frac{\gamma}{2}) \end{bmatrix}$$

Pauli-Y                 CNOT gate                 Pauli-X                 Pauli-Z

(b) Matrix representation for the Quantum gates applied in the VQC

Figure 4: *A framework of variational quantum circuit.*

### 4.3 THE FRAMEWORK OF VARIATIONAL QUANTUM CIRCUIT

The framework of VQC is shown in Figure 4 (a), where 4 qubit wires are taken into account, and the CNOT gates aim at mutually entangling the channels such that $|x_1\rangle$, $|x_2\rangle$, $|x_3\rangle$ and $|x_4\rangle$ lie in the same entanglement state. The Pauli-X, Y, Z gates $R_X(\cdot)$, $R_Y(\cdot)$ and $R_Z(\cdot)$ with learnable parameters $(\alpha_1, \beta_1, \gamma_1)$, $(\alpha_2, \beta_2, \gamma_2)$, $(\alpha_3, \beta_3, \gamma_3)$, $(\alpha_4, \beta_4, \gamma_4)$ are built to set up the learnable part. Being similar to the unitary operators of $R_Y(\alpha)$, $R_X(\beta)$ and $R_Z(\gamma)$, which are defined in Figure 4 (b), are separately associated with the rotations along X-axis and Z-axis by the given angles of $\beta$ and $\gamma$. Besides, the quantum circuits in the dash square can be repeatedly copied to compose a deeper architecture. The outputs of VQC are connected to the measurement which projects the quantum states into a certain quantum basis that becomes a classical scalar $z_i$.

As for the end-to-end training paradigm for QTN-VQC, the learnable parameters come from the VQC and TTN models, and they should be updated by applying the back-propagation algorithm based on the Adam optimizer. Given $D$ qubits and $H$ depths, there are totally $3DH$ trainable parameters for VQC. Consequently, there are $\sum_{k=1}^{K} r_{k-1} r_k I_k J_k + 3DH$ parameters for QTN-VQC. On the other hand, the Dense-VQC model possesses more model parameters than QTN-VQC ($\prod_{k=1}^{K} r_{k-1} r_k I_k J_k + 3DH$ vs. $\sum_{k=1}^{K} r_{k-1} r_k I_k J_k + 3DH$).

## 5 CHARACTERIZING REPRESENTATION POWER OF QTN-VQC

This section focuses on analyzing the representation power of QTN-VQC. As shown in Figure 5, given $d$ qubits and a target quantum state $|\mathbf{z}\rangle = \otimes_{d=1}^{D} |z_d\rangle$, since $H_{\boldsymbol{\theta}}$ is known as a linear operator and $\mathcal{T}_{\mathbf{x}}$ is defined as a definite mapping from input $\mathbf{x}$ to the unitary matrix $U_{\mathbf{x}}$, the representation power of QTN-VQC is determined by how TTN can approximate the classical vector $\mathcal{T}_{\mathbf{x}}^{-1}(H_{\boldsymbol{\theta}}^{-1}|\mathbf{z}\rangle)$. To understand the expressiveness of TTN, we first start with the discussion on the expressive capability of Dense-VQC (a dense layer is taken for dimension reduction) and then generalize it to QTN-VQC. Based on the universal approximation theorem (Cybenko, 1989; Barron, 1994) for a feed-forward neural network, we derive the following theorem as:

**Theorem 2.** *Given a target vector $\mathcal{T}_{\boldsymbol{x}}^{-1}(H_{\boldsymbol{\theta}}^{-1}|\mathbf{z}\rangle)$, there exists a feed-forward neural network $f_{dense}$ with a dense layer connecting to $D$ qubits, then*

$$|| \mathcal{T}_{\boldsymbol{x}}^{-1}(H_{\boldsymbol{\theta}}^{-1}|\mathbf{z}\rangle) - f_{dense}(\boldsymbol{y}) ||_1 \leq \frac{C}{\sqrt{D}}, \tag{9}$$

*where the activation function $\tanh(\cdot)$ is imposed upon the dense layer, and $C$ is a constant associated with the target vector $\mathcal{T}_{\boldsymbol{x}}^{-1}(H_{\boldsymbol{\theta}}^{-1}|\mathbf{z}\rangle)$.*

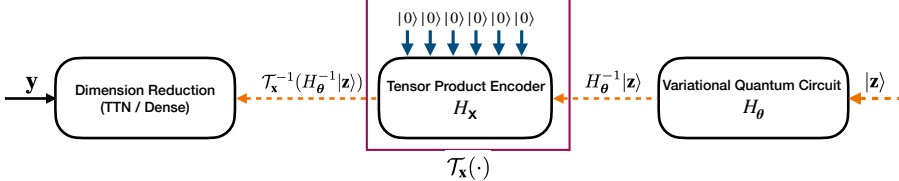

Figure 5: *An illustration of analyzing the representation power of QTN-VQC.*

Since TTN is a compact TT representation of a dense layer, by modifying Theorem 2 for TTN, we can also derive the upper bound on the approximation error as follows:

**Theorem 3.** *Given a target vector $\mathcal{T}_{\boldsymbol{x}}^{-1}(H_{\boldsymbol{\theta}}^{-1}|\boldsymbol{z}\rangle)$, there exists a TTN, denoted as $f_{TTN}$, with a TT layer connecting to $D$ qubits, then*

$$|| \, \mathcal{T}_{\boldsymbol{x}}^{-1}(H_{\boldsymbol{\theta}}^{-1}|\boldsymbol{z}\rangle) - f_{TTN}(\boldsymbol{y}) \, ||_1 \le \prod_{k=1}^{K} \frac{C}{\sqrt{D_k}}, \qquad (10)$$

*where $\prod_{k=1}^{K} D_k = D$, the Sigmoid activation function is imposed upon the TTN model, $K$ denotes the multi-dimensional order, $C$ is a constant associated with the target vector $\mathcal{T}_{\boldsymbol{x}}^{-1}(H_{\boldsymbol{\theta}}^{-1}|\boldsymbol{z}\rangle)$.*

Comparing the two upper bounds, it is observed that TTN can attain an identical upper bound as the dense layer on the approximation error because $\prod_{k=1}^{K} D_k = D$. That implies that TTN can at least maintain the representation power of a dense layer. Besides, the number of qubits $D$ is a key factor determining the upper bound on the approximation error. However, $D$ is a small fixed number on a NISQ device, and a larger number of qubits $D$ is expected to further improve the representation power of QTN-VQC. However, the computational costs of classical simulation may grow exponentially with the increasing number of qubits, and a small number of qubits have to be considered in practice.

## 6 EXPERIMENTS AND RESULTS

### 6.1 EXPERIMENTAL SETUPS

We assess our QTN-VQC based end-to-end learning system on the standard MNIST. MNIST is a dataset for the task of 10 digit classification, where there are 50000 and 10000 $28 \times 28$ image data assigned for training and testing, respectively. The full MNIST dataset is challenging for quantum machine learning algorithms, and many works only consider 2-digit classification on the MNIST task (Wang et al., 2021; Chen et al., 2020a). Moreover, the image data are separately reshaped into 784 dimensional input vectors. Dense-VQC and PCA-VQC are taken as our experimental baselines to compare with our QTN-VQC model. Dense-VQC denotes that a dense layer is used for dimension reduction, and PCA-VQC refers to using principal component analysis (PCA) to extract low-dimensional features before training the VQC parameters.

As for the experiments of QTN-VQC, the image data are reshaped into 3-order $7 \times 16 \times 7$ tensors. We set small TT-ranks as $\{1, 2, 2, 1\}$ to reduce the computational cost of TTN. the image data are represented as the TT format according to Eq. (3) before going through the TTN model. Since 8 qubits are used for the quantum encoding, the output of TTN needs to configure the tensor format as $2 \times 2 \times 2$, which results in 8 dimensional output vectors. Besides, the model parameters of QTN-VQC are randomly initialized based on the Gaussian distribution, and the back-propagation algorithm is applied to train the models. The Sigmoid function is utilized for the hidden layers of TTN.

To be consistent with QTN-VQC, the weight of the dense layer for Dense-VQC is configured as the shape of $784 \times 8$. Although Dense-VQC is a hybrid classical-quantum model, the training process of Dense-VQC can also be set as an end-to-end pipeline and the weights of the dense layer are updated during the training stage. The Sigmoid function is used for the dense layer. On the other hand, PCA

is employed to reduce the feature dimension to $8$, and the resulting low-dimensional features are further encoded into quantum states. Consequently, PCA-VQC admits the VQC parameters solely to be updated during the training stage. A standard AlexNet (Iandola et al., 2016) is employed to constitute an AlexNet-VQC to compare the performance.

Moreover, 6 VQC layers are constructed to form a deep model, and the outputs of the VQC model are connected to $10$ classes with a non-trainable matrix. The back-propagation algorithm based on the Adam optimizer with a learning rate of $0.001$ is employed for the model training. The loss of cross-entropy (CE) is utilized as the objective function during the training stage, and it is also taken as the metric to evaluate the model performance. We leverage the tools of Pennylane (Bergholm et al., 2018) and PyTorch (Paszke et al., 2019) to simulate the model performance. In particular, we separately simulate the model performance with noiseless quantum circuits and noisy quantum circuits corrupted by quantum noises from IBM quantum machines.

## 6.2 Experimental Results of Noiseless Quantum Circuit

Table 1 shows the final results of the models on the test dataset. QTN-VQC owns much fewer model parameters than Dense-VQC ($328$ vs. $6416$) and attains even higher classification accuracy than Dense-VQC ($91.43\%$ vs. $88.54\%$) and lower loss values than Dense-VQC ($0.3090$ vs. $0.4132$). However, PCA-VQC with $144$ trainable VQC parameters attains the worst performance by all metrics, which implies that a trainable quantum embedding is of significance to boost experimental performance. Although our empirical results cannot reach the state-of-the-art classification performance of classical ML algorithms, our empirical results demonstrate the advantages of QTN-VQC over the PCA-VQC and Dense-VQC counterparts. With the development of more powerful quantum devices supporting more qubits, the representation power of QTN-VQC can be improved and better experimental results could be attained. Moreover, AlexNet-VQC achieves better results than QTN-VQC ($92.81\% vs. 91.43\%$), but it involves more model parameters than QTN-VQC.

Table 1: Empirical results on the MNIST test dataset under the noiseless quantum circuit setting.

| Models | Params | CE | Acc (%). |
|---|---|---|---|
| PCA-VQC | 144 | 0.5877 | $82.48 \pm 1.02$ |
| Dense-VQC | 6416 | 0.4132 | $88.54 \pm 0.73$ |
| AlexNet-VQC | $3.25 \times 10^6$ | 0.2562 | $92.81 \pm 0.47$ |
| QTN-VQC | 328 | **0.3090** | $\mathbf{91.43 \pm 0.51}$ |

## 6.3 Experimental Results of Noisy Quantum Circuit

To empirically validate the effectiveness of our proposed VQC algorithm, we proceed with the simulation of the practical experiments with noisy quantum circuits. More specifically, we follow an established noisy circuit experiment with the NISQ device suggested by (Chen et al., 2020b). One major advantage of the setups is to observe the robustness and preserve the quantum advantages of a deployed VQC with physical settings being close to quantum processing unit (QPU) experiments without an executive queuing time. As for the detailed setup, we first use an IBM Q 20-qubit machine to collect channel noise in the real scenario for a deployed VQC and upload the machine noise into our Pennylane-Qiskit simulator (denoted as $\text{Acc}_{q20}$. We provide a depolarizing noisy circuit simulation (denoted as $\text{Acc}_{depo}$) based on a depolarizing channel attained from (Nielsen & Chuang, 2010) with a noise level of $0.1$. As shown in Table 2, the quantum noise brings about the performance degradation of all models, but our proposed QTN-VQC consistently outperforms PCA-VQC and Dense-VQC in the condition of noisy quantum circuits. In particular, QTN-VQC can even outperform the AlexNet-VQC counterpart in noisy circuit conditions.

## 6.4 Further Discussions

The above experimental results show the advantages of QTN-VQC over Dense-VQC and PCA-VQC in the scenarios with noiseless and noisy quantum circuits. Next, we will further discuss the representation power of QTN-VQC based on two factors: (1) the activation function used in TTN; (2) the number of qubits.

Table 2: Empirical results on the MNIST test dataset under the noisy quantum circuit setting.

| Models | Params | $\text{Acc}_{q20}$ (%) | $\text{Acc}_{depo}$ (%) |
|---|---|---|---|
| PCA-VQC | 144 | $81.23 \pm 1.34$ | $83.12 \pm 1.17$ |
| Dense-VQC | 6416 | $84.55 \pm 1.22$ | $86.09 \pm 1.04$ |
| AlexNet-VQC | $3.25 \times 10^6$ | $87.46 \pm 1.34$ | $87.86 \pm 1.08$ |
| QTN-VQC | 328 | $\mathbf{88.12 \pm 1.09}$ | $\mathbf{89.32 \pm 1.07}$ |

### 6.4.1 THE ACTIVATION FUNCTION USED IN TTN

Table 3 compares the results of QTN-VQC based on different activation functions. Our simulation on noiseless quantum circuits shows that the non-linear activation functions can bring more performance gain than a linear one, but the Sigmoid function attains a better performance than the Tanh and ReLU counterparts in our experiments. Our experiments also correspond to the universal approximation theory for QTN-VQC in Theorem 3.

Table 3: Comparing performance of QTN-VQC with and without activation function.

| Models | CE | Acc (%). |
|---|---|---|
| QTN-VQC (Linear) | 0.4958 | $86.16 \pm 0.65$ |
| QTN-VQC (Tanh) | 0.4792 | $87.12 \pm 0.51$ |
| QTN-VQC (ReLU) | 0.3764 | $89.56 \pm 0.49$ |
| QTN-VQC (Sigmoid) | $\mathbf{0.3090}$ | $\mathbf{91.43 \pm 0.54}$ |

### 6.4.2 THE NUMBER OF QUBITS

Finally, we investigate the effects of the number of qubits on the performance of QTN-VQC by increasing the qubits from 8 to 12 and 16. Accordingly, the output of TTN is configured as a tensor format of $2 \times 3 \times 2$, and the model size is increased from 328 to 464 and 600, respectively. Our experiments show that the baseline performance of QTN-VQC can be further improved by increasing the number of qubits, which implies that more qubits are likely to possess higher accuracy.

Table 4: Comparing performance of QTN-VQC with fewer qubits.

| Models | Params | CE | Acc (%) |
|---|---|---|---|
| QTN-VQC (8 qubits) | 328 | 0.3090 | $91.43 \pm 0.51$ |
| QTN-VQC (12 qubits) | 464 | 0.2679 | $92.36 \pm 0.62$ |
| QTN-VQC (16 qubits) | 600 | $\mathbf{0.2355}$ | $\mathbf{92.98 \pm 0.52}$ |

## 7 CONCLUSIONS

This work proposes a genuine end-to-end learning framework for quantum neural networks based on QTN-VQC. QTN consists of a TTN for dimension reduction and a TPE framework for generating quantum embedding. The TTN model is a compact representation of a dense layer to classically simulate quantum machine learning algorithms. Our theorem on the representation of QTN-VQC shows that the number of qubits is inversely related to the approximation error of QTN-VQC and the non-linear activation plays an important role. Our experiments compare our proposed QTN-VQC with Res-VQC, Dense-VQC, and PCA-VQC. Our simulated results demonstrate that QTN-VQC obtains better experimental performance than Dense-VQC and PCA-VQC with both noiseless and noisy quantum circuits, and it achieves marginally worse performance than AlexNet-VQC. Besides, our results justify our theorem on the representation power of QTN-VQC.

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

## A APPENDIX

The section of appendix includes the proofs for Theorem 2 and Theorem 3.

### A.1 PROOF FOR THEOREM 2

*Proof.* Theorem 2 is derived from the modification of the universal approximation theory proposed by Barron (1994); Cybenko (1989). The universal approximation theory is shown in Lemma 1, which suggests that a feed-forward neural network with $d$ neurons can approximate any continuous function with arbitrarily small $\epsilon$.

**Lemma 1.** *Given a continuous target function $\hat{f} : \mathbb{R}^I \to \mathbb{R}$, we can employ a 2-layer neural network with a non-linear activation $f : \mathbb{R}^I \to \mathbb{R}$, such that*

$$|\hat{f} - f| \leq \frac{1}{\sqrt{D}} C_{\hat{f}}, \tag{11}$$

*where $J$ denotes the number of neurons, and $C_{\hat{f}}$ is a constant associated with $\hat{f}$. In particular, for $r \geq 1$, $C_f$ satisfies the following condition as:*

$$||\hat{f}||_\infty + \sum_{k, 1 \leq \frac{k(k-1)}{2} \leq r} ||\mathcal{D}^k \hat{f}||_\infty \leq C_{\hat{f}}, \tag{12}$$

*where $\mathcal{D}^k \hat{f} = \left[ \nabla \hat{f}, \nabla^2 \hat{f}, ..., \nabla^k \hat{f} \right]^T$.*

To associate Lemma 1 with our Theorem 2, the target function is replaced with the target vector $H_X^{-1} H_\theta^{-1} |\mathbf{z}\rangle$, then there is a neural network with a dense layer connected to $D$ qubits such that

$$||H_X^{-1} H_\theta^{-1} |\mathbf{z}\rangle - f_{dense}(\mathbf{y})||_1 \leq \frac{1}{\sqrt{D}} C, \tag{13}$$

where $C$ is related to the target vector $H_X^{-1} H_\theta^{-1} |\mathbf{z}\rangle$. □

### A.2 PROOF FOR THEOREM 3

*Proof.* Assume that $\mathcal{X} = f_{TTN}(\mathbf{y})$, $\hat{\mathcal{X}} = H_X^{-1} H_\theta^{-1} |\mathbf{z}\rangle$ and the TT decomposition of target vector is $\{\hat{\mathcal{X}}_1, \hat{\mathcal{X}}_2, ..., \hat{\mathcal{X}}_K\}$, then we obtain

$$||H_X^{-1} H_\theta^{-1} |\mathbf{z}\rangle - f_{TTN}(\mathbf{y})||_1 = ||\hat{\mathcal{X}} - \mathcal{X}||_1 \leq \prod_{k=1}^{K} ||\hat{\mathcal{X}}_k - \mathcal{X}_k||_1 \tag{14}$$

On the other hand, we denote $\text{vec}(\mathcal{Y}_k)$ and $\text{vec}(\mathcal{X}_k)$ as the vectorization of the tensors $\mathcal{Y}_k$ and $\mathcal{X}_k$, respectively. We also define $\prod_{k=1}^{K} I_k = I$, $\mathcal{W}_k \in \mathbb{R}^{D_k \times I_k \times r_{k-1} \times r_k}$ as the TTN parameters, and also define $\mathbf{W}_k \in \mathbb{R}^{I_k \times r_{k-1} r_k D_k}$ as the matricization of $\mathcal{W}_k$. Moreover, $\sigma$ refers to a non-linear activation function.

Since $\text{vec}(\mathcal{X}_k) = \sigma(\mathbf{W}^T \text{vec}(\mathcal{Y}_k))$ that corresponds to a dense layer, we can obtain that

$$||\hat{\mathcal{X}}_k - \mathcal{X}_k||_1 \leq \frac{1}{\sqrt{D_k}} C. \tag{15}$$

In sum, we can further obtain

$$||H_X^{-1} H_\theta^{-1} |\mathbf{z}\rangle - f_{TTN}(\mathbf{y})||_1 \leq \prod_{k=1}^{K} ||\hat{\mathcal{X}}_k - \mathcal{X}_k||_1 \leq \prod_{k=1}^{K} \frac{1}{\sqrt{D_k}} C, \tag{16}$$

where $\prod_{k=1}^{K} D_k = D$. □

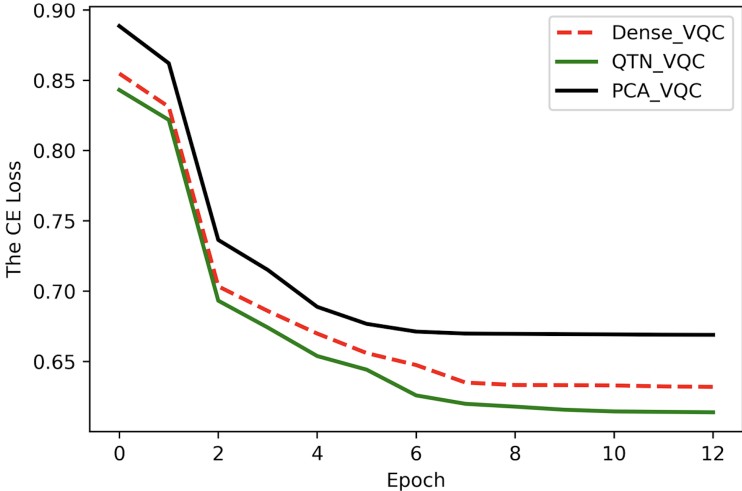

Figure 6: *A comparison of convergence rates for different models.*

## B APPENDIX

This section includes additional experimental simulations. First, we assess the settings of TT-ranks, and then we compare the convergence rates of QTN-VQC and Dense-VQC in the experiments.

### B.1 EXPERIMENTS ON TT-RANKS FOR QTN-VQC

Table 5 corresponds to the experiments of QTN-VQC with 8 qubits and the Sigmoid function. The empirical results suggest that the larger TT-ranks cannot result in better results than the smaller ones. The main reason is that the TT-ranks can correspond to a manifold, and there may potentially exist an optimal manifold with smaller TT-ranks that corresponds to the best performance.

Table 5: Comparing performance of different TT-ranks for QTN-VQC

| TT-ranks | Params | CE | Acc (%) |
|---|---|---|---|
| $\{1, 2, 2, 1\}$ | 328 | 0.3090 | $91.43 \pm 0.51$ |
| $\{1, 4, 4, 1\}$ | 768 | 0.3082 | $91.46 \pm 0.53$ |
| $\{1, 6, 6, 1\}$ | 1464 | 0.3079 | $91.47 \pm 0.52$ |

### B.2 A COMPARISON OF CONVERGENCE RATES

Next, we analyze the computational complexity for TTN for QTN-VQC. In more detail, given the TT-ranks $\{r_1, r_2, ..., r_K\}$, a multi-dimensional tensor $\mathcal{W}$ is factorized into several $K$-order tensors $\mathcal{W}_k \in \mathbb{R}^{r_{k-1} \times I_k \times J_k \times r_k}$, the computational complexity of the feed-forward process is in the scale of $\mathcal{O}(K \max_k I_k \max_k J_k (\max_k r_k)^K)$. In contrast, the computational overhead for a dense layer is in the scale of $\mathcal{O}(\prod_k I_k \prod_k J_k)$. It means that smaller TT-ranks can reduce the computational cost for QTN-VQC, which explains that smaller TT-ranks $\{1, 2, 2, 1\}$ is configured in our experiments of QTN-VQC.

Empirically, we compare the convergence rates of different models on the test data in our experiments. In our experimental settings with the Tanh activation function and 8 qubits, the QTN-VQC model consistently attains a faster convergence rate than the Dense-VQC and PCA-VQC counterparts. Moreover, Table 6 compares the absolute running time of QTN-VQC with Dense-VQC and AlexNet-VQC. Since our experiments are conducted on the same GPUs and CPUs, the training time of all models can be comparable. Our evaluation shows that QTN-VQC is marginally slower than Dense-VQC, but it is much faster than AlexNet-VQC.

Table 6: Comparing performance of different TT-ranks for QTN-VQC

| Models | Dense-VQC | AlexNet-VQC | QTN-VQC |
|---|---|---|---|
| Time/epochs (mins) | 58 | 75 | 61 |

## C  Experiments of Labeled Faces in the Wild (LFW)

### C.1  Experimental setups

The LFW is a dataset for the task of unconstrained face recognition, which is composed of 13000 images with the shape of $[154, 154, 3]$. The shape of We randomly split all the datasets into 11000 training data, 2000 test data. 16 qubits are used for VQC, and the shape of the input tensor is set as $22 \times 147 \times 22$. The other settings are kept the same as the configurations for the MNIST task.

### C.2  Experimental results

Table 6 presents the simulation results under the noiseless quantum circuit condition, while Table 7 demonstrates the empirical results in the setting of noisy quantum circuits. The QTN-VQC outperforms the Dense-VQC counterpart (92.15 vs. 91.27), and it owns much fewer model parameters (2816 vs. $1.1 \times 10^6$). Although

The experimental results on the LFW dataset also highlight the advantages of QTN-VQC in terms of fewer model parameters and better empirical performance.

Table 7: Simulation results on the LFW test dataset under a noiseless circuit condition.

| Models | Params | CE | Acc (%) |
|---|---|---|---|
| Dense-VQC | $1.1 \times 10^6$ | 0.3011 | $91.27 \pm 0.25$ |
| AlexNet-VQC | $2.3 \times 10^7$ | 0.2875 | $93.21 \pm 0.36$ |
| QTN-VQC | 2816 | **0.2910** | **$92.15 \pm 0.43$** |

Table 8: Empirical results on the LFW test dataset under the noisy quantum circuit setting.

| Models | Params | $Acc_{q20}$ (%) | $Acc_{depo}$ (%) |
|---|---|---|---|
| Dense-VQC | $1.1 \times 10^6$ | $88.65 \pm 1.22$ | $87.23 \pm 1.04$ |
| AlexNet-VQC | $2.3 \times 10^7$ | $89.76 \pm 1.34$ | $88.66 \pm 1.08$ |
| QTN-VQC | 2816 | **$89.93 \pm 1.09$** | **$89.64 \pm 1.07$** |

