# OpenReview forum: "QTN-VQC: An End-to-End Learning Framework for Quantum Neural Networks"
_ICLR.cc/2022/Conference — ICLR 2022 Submitted_

### Official Review · Reviewer_Awxc · 2021-10-20

**Correctness:** 3
**Technical Novelty And Significance:** 2
**Empirical Novelty And Significance:** 2
**Recommendation:** 5
**Confidence:** 3

**Main Review:**

Pros:

1. The paper tackles a real limitation for QNN by replacing the dense NN with TTN and enabling end-to-end training.
2. The author provides codes for reproducing, which is great.

Cons:

1. The related work is insufficient. Since the TTN is the key proposal of this paper, it should be explained more in the related works. For example, as an existence module, how it works in the convolutional neural network and recurrent neural networks. What is the difference between this paper and previous literature, excluding the application difference?
2. The ablation is highly insufficient. Especially, in table 3/4, only 1 non-linear activation and two # qubits options are evaluated.
3. The technical contribution seems limited. Since the authors do not touch the QNN and only use an existing module TTN to replace the previous dense layer for pre-processing high-dimensional input.

**Summary Of The Paper:**

Summary:

This paper is dedicated to designing an end-to-end learning framework for quantum neural networks. The authors design a novel quantum tensor network for dimension reduction and quantum embeddings generation, since it is quite related to the practical usage in real-world applications where only a small number of qubits could be supported on available NISQ computers at this moment. The key contribution of this paper is to leverage a tensor train network (TTN) to replace the dense layer for dimension reduction, which enables an end-to-end training process fully conducted in a quantum computer.

**Summary Of The Review:**

Due to the limited technical contribution/novelty and insufficient experiments, I tend to borderline reject the paper.

---

> ### Author Response · Authors · 2021-11-19
> **Reply to Reviewer Awxc**
>
> **Q1.** The related work is insufficient. Since the TTN is the key proposal of this paper, it should be explained more in the related works. For example, as an existence module, how it works in the convolutional neural network and recurrent neural networks. What is the difference between this paper and previous literature, excluding the application difference?
>
> **Ans:** Thanks for the reviewer’s comment on this aspect. We have strengthened the descriptions in the related work for the usage difference of TTN. In fact, in this work, we leverage TTN to realize a tensor network for a QTN-VQC based end-to-end learning framework for quantum neural networks. In other words, this work does not simply compress a dense layer with TTN, but we implement a quantum tensor network to classically simulate quantum machine learning algorithms.
>
> ***
>
> **Q2.** The ablation is highly insufficient. Especially, in table 3/4, only 1 non-linear activation and two # qubits options are evaluated.
>
> **Ans:** Thanks for pointing that out. As shown in Table 3, we have attempted different activation functions (Tanh, ReLU, and Sigmoid) in our experiments, and we observe that the use of the Sigmoid function can further improve the baseline performance.
>
> ***
>
> **Q3.** The technical contribution seems limited because the authors do not touch the QNN and only use an existing module TTN to replace the previous dense layer for pre-processing high-dimensional input.
>
> **Ans:** Thanks again for the summarization of this work. The main contribution of this work is to implement a genuine quantum tensor network for an end-to-end learning paradigm to classically simulate quantum machine learning algorithms. Besides, we also conducted a theoretical analysis on the representation power of QTN-VQC, where we bridge the connection between the performance and the number of qubits. For sure, we intend to design more complicated parametric quantum circuits, such as 2D tensor networks and VQC with quantum error correction, in our future work. But this work concentrates on the implementation of a simple quantum tensor network and compares the performance with the hybrid classical-quantum models.

---

### Official Review · Reviewer_8JcS · 2021-10-21

**Correctness:** 3
**Technical Novelty And Significance:** 2
**Empirical Novelty And Significance:** 2
**Recommendation:** 3
**Confidence:** 3

**Main Review:**

Below, I describe the following issues of the paper:
-	The main contribution is the idea of using TTN for compressing input data, which seems like a trivial approach.
-	Moreover, the selected TTN structure seems not to be optimal for image treatment. It is well known that, to obtain best compression rates of images with less distortion, data should be reshaped in order to collect small patches and group rows and columns indices. More specifically, given a IxJ image, we can obtain a good TT compression if we make the following transforms IxJ -> I1xI2x…xINxJ1xJ2x…xJN -> I1xJ1xI2xJ2x…INxJN -> (I1J1)x(I2J2)x…x(INJN).  I would suggest trying those optimal data transformations before applying the TT model which will contribute to obtain better classification accuracies.
-	Only MNIST dataset was explored. It would be needed that other types of datasets were explored and analyzed.
-	I found the TT-ranks {1,2,2,1} too small. It would be usefult to have a visualization of the input data after compression to assess the introduced distortion. Also, it would be useful to report results with larger TT-ranks too.
-	Provided performance are poor compared to simple classical ML algorithms. For example, using a very simple Logistic Regression classifier can provide testing accuracy around 92%.
-	In the definitions section, it should be clear that K is the number of input entries and d the number of qbits.
-	Theorem 1 is trivial and not needed. In fact, the proposed embedding of input data was already used in many papers, see for example [Stoudenmire, 2016].
References:
[Stoudenmire, 2016] Stoudenmire, E. M., & Schwab, D. J. (2016). Supervised Learning with Quantum-Inspired Tensor Networks. arXiv, 1605, arXiv:1605.05775.

**Summary Of The Paper:**

As it was already studied in the literature, one way to implement a machine learning classifier in quantum computers is to convert classical input data like images into quantum states that are fed to a Variational Quantum Circuit (VQC). However, current quantum computers have a small number of qbits, which requires to perform a dimensionally reduction of the input datasets as a preprocessing step. In this paper, the authors propose to use data compression based on the Tensor Train Network (TTN) model, which is a very well-known compression technique. They compare TTN compression against a straightforward dense layer and the PCA-based dimension reduction techniques. The paper includes experimental results on MNIST dataset by simulating quantum circuits with up to 8qbits for noisy and noiseless scenarios.

**Summary Of The Review:**

Strengths:
- Quantum Machine Learning is a very important and growing field of research. This paper shows one potential way to use quantum computers for machine learning.

Weaknesses:
- The contribution of the paper is limited (see comment below).
- Experimental results are also limited (see comments below)

---

> ### Author Response · Authors · 2021-11-19
> **Reply to Reviewer 8JcS**
>
> **Q1.** Moreover, the selected TTN structure seems not to be optimal for image treatment. It is well known that to obtain the best compression rates of images with less distortion, data should be reshaped in order to collect small patches and group rows and columns indices. More specifically, given a IxJ image, we can obtain a good TT compression if we make the following transforms IxJ -> I1xI2x…xINxJ1xJ2x…xJN -> I1xJ1xI2xJ2x…INxJN -> (I1J1)x(I2J2)x…x(INJN). I would suggest trying those optimal data transformations before applying the TT model which will contribute to obtaining better classification accuracies.
>
> **Ans:** We thank the reviewer’s suggestions on the experiments. However, in this work, we only leverage the most simple TTN to implement a tensor network for dimension reduction. In doing so, the computational cost of TTN can be reduced to the minimum. Besides, the use of TTN admits a QTN-VQC which allows an end-to-end learning paradigm for quantum neural networks (QNN). Moreover, we intend to use a standard ResNet18 model for dimension reduction followed by VQC. We found that the performance can even attain better results than QTN-VQC, but ResNet-VQC is more sensitive to quantum noise effects than QTN-VQC, which is shown in our new experiments.
>
> ***
>
> **Q2.** Only the MNIST dataset was explored. It would be needed that other types of datasets were explored and analyzed.
>
> **Ans:** As suggested by the reviewer, we have attempted another experimental task based on the LFW dataset, which is shown in Appendix C. The task on the LFW dataset refers to a more challenging unconstrained face recognition, where we attempt 16 qubits and Sigmoid activation function in the experiments under both the noiseless and noisy quantum circuit settings. The related experimental results corroborate our theoretical analysis.
>
> ***
>
> **Q3.** I found the TT-ranks {1,2,2,1} too small. It would be useful to have a visualization of the input data after compression to assess the introduced distortion. Also, it would be useful to report results with larger TT-ranks too.
>
> **Ans:** Thanks for the reviewer’s suggestion. Since we employ the most simple TTN to realize a tensor network for dimension reduction because of the concern of computational complexity, we deliberately consider the small TT-ranks in our original work. But we follow the reviewer’s comment to try larger TT-ranks and show the results in the Appendix.
>
> ***
>
> **Q4.** Provided performance is poor compared with simple classical ML algorithms. For example, using a very simple Logistic Regression classifier can provide testing accuracy of around 92%.
>
> **Ans:** Thanks for pointing that out. We further boost the performance by comparing different activation functions (Table 3) and attempting more qubits (Table 4) in the experiments. However, the new results are still below the state-of-the-art results by classical neural networks. Our results are consistent with the website (https://www.tensorflow.org/quantum/tutorials/mnist) which demonstrates the MNIST results by using TensorFlow quantum, where it is claimed that quantum neural networks are difficult to beat classical neural networks in dealing with classical datasets. However, our theoretical analysis in Theorem 2 suggests that the increase of qubits can boost the performance, which can close the gap between the quantum and classical neural networks for classical data.
>
> ***
>
> **Q5.** Theorem 1 is trivial and not needed. In fact, the proposed embedding of input data was already used in many papers, see for example [Stoudenmire, 2016]. References: [Stoudenmire, 2016] Stoudenmire, E. M., & Schwab, D. J. (2016). Supervised Learning with Quantum-Inspired Tensor Networks. arXiv, 1605, arXiv:1605.05775.
>
> **Ans:** Thank you for pointing that out. Although [Stoudenmire, 2016] provides a quantum encoding plan, in this work, we delicately design quantum circuits to implement the quantum encoding which is associated with Theorem 1. Thus, Theorem 1 highlights the composition of quantum circuits applied in this work.

---

> > ### Comment · Reviewer_8JcS · 2021-11-19
> > **Thank you for the responses**
> >
> > I would like to thank authors for providing responses to my questions.
> > Unfortunately, I keep my previous score.
> > I would encourage the authors to improve the work for a future publication

---

### Official Review · Reviewer_JaBi · 2021-10-31

**Correctness:** 3
**Technical Novelty And Significance:** 2
**Empirical Novelty And Significance:** 2
**Recommendation:** 5
**Confidence:** 4

**Main Review:**

Strength:
- The idea of combining QTT with VQC is interesting, which is helpful for quantum neural network.
- The paper presents quantum neural network in a nicely understandable language for machine learning researchers although there are still some minor mistakes or confusion.

Weakness:
- The novelty of this paper is fair.  This paper combines QTT and VQC into one framework. TT has been applied to fully connected layer for model compression by many existing works.  VQC is well known quantum machine learning model.  TPE is a simple method to encode classical data as quantum state, which is also proposed by quantum inspired neural network (NeurIPS 2016).

- The experiment validation is done on MINIST dataset, however, the performance is far from the state-of-the-art performance by DNN, which cannot show how useful of quantum neural network.  Since quantum computing is based on simulation, can you increase the number of qubits and obtain better performance on this simply MINIST task?

- It shows the number of parameters are much smaller by using QTT.  But how about computational cost and convergence rate when comparing with dense network?

- Other experiments on larger datasets like CIFAR is expected, which is related to the scalability of the propose algorithm.

- When TT is used to approximate the fully connected weights, the gradient exploding and diminishing often occur due to many of core tensors.  But the paper has not presented any algorithm related study and discussion.

- Input data x is reshaped into a k-order tensor, and then it is represented as TT format. However, this is the procedure of TT decomposition of data x, which is still challenging task and computational expensive.

- In the Theorem 3, the upper bound is not related to TT-ranks, which seems not reasonable. Let us assume TTN with tt-ranks are all 1, can we achieve same upper bound as dense layer?

Minor comments:
- Eq. (2) is confusing, it should be that v is m-dimensional vector, and each v_i is 2-dimensional vector.
- Typo in the line below eq.(4), the core tensor is X rather than W.
- Eq. (6) and (9) are same. It is not necessary to be written twice.
- The comparison of model parameters between QTN-VQC and Dense-TPE-VQC in the last line of Sec. 4 is also wrong.


**Summary Of The Paper:**

This work proposes an end-to-end learning framework TTN-VQC for quantum neural networks. The main problem is that current quantum computer cannot handle large number of qubits.  The idea of this paper is applying QTT to perform dimensional reduction followed by quantum encoding TPE, which can be combined with existing quantum neural network (VQC) as an end-to-end learning model.

**Summary Of The Review:**

The paper is an improvement of quantum neural network algorithm.  The idea is simply but interesting for machine learning community.  However, novelty is fair, and the experimental evaluation is less convincing. Some important problems like computational cost, gradient exploding are not well discussed.

---

> ### Author Response · Authors · 2021-11-19
> **Reply to Reviewer JaBi (1/2)**
>
> **Q1.** The novelty of this paper is fair. This paper combines QTT and VQC into one framework. TT has been applied to fully connected layers for model compression by many existing works. VQC is a well-known quantum machine learning model. TPE is a simple method to encode classical data as a quantum state, which is also proposed by quantum-inspired neural networks (NeurIPS 2016).
>
> **Ans:** In this work, the main idea is not a simple application of TT to compress an FC layer. In fact, we leverage a parametric TTN for quantum embedding generation such that an end-to-end learning framework can be built for quantum neural networks (QNN).
>
> In particular, the QTN-VQC model with the TTN based quantum embedding can be regarded as the classical simulation of the corresponding quantum machine learning. Moreover, although our TPE is mathematically matched with the math expression in [Stoudenmire and Schwab, 2016], we set up the connection between the math and quantum gates as shown in Theorem 1. More significantly, **we theoretically analyze the representation power** of QTN-VQC as an end-to-end learning framework for QNN, which highlights the importance of a parametric TTN for quantum embedding and **bridges an essential relationship between performance and the number of qubits.**
>
> ***
>
> **Q2.** The experimental validation is done on the MNIST dataset, however, the performance is far from the state-of-the-art performance by DNN, which cannot show how useful a quantum neural network is. Since quantum computing is based on simulation, can you increase the number of qubits and obtain better performance on this simple MNIST task?
>
> **Ans:** Thanks for the reviewer’s comments. In our updated paper, we further improve the empirical performance by increasing the number of qubits (Table 4) and comparing more activation functions (Table 3).
>
> Although our quantum machine learning results are still below the state-of-the-art performance, we have found that more qubits can contribute to the performance gain, which corresponds to our theoretical analysis in Theorem 2 for QTN-VQC.
> Besides, the website on (https://www.tensorflow.org/quantum/tutorials/mnist) for the tutorial of Tensorflow Quantum shows similar simulation outcomes with ours, and it is claimed that it is difficult for QNN to beat classical neural networks when dealing with classical data. Additionally, our work also shows that a classical simulation based on tensor networks can narrow the gap between quantum and classical models for the classical data.
>
> ***
>
> **Q3.** It shows the number of parameters is much smaller by using QTT. But how about computational cost and convergence rate when comparing a dense network?
>
> **Ans:** Thank you for pointing out the experimental factors.
>
> As discussed in Section 4.1, the computational costs of TTN and a dense layer are on the same scale, and the given tensor-train ranks for TTN can proportionally affect the related computational complexity and convergence rate. In our experiments, the convergence rates can be at the same pace if we delicately assign the learning rates for TTN and a dense layer.
>
> ***
>
> **Q4.** Other experiments on larger datasets like CIFAR are expected, which is related to the scalability of the proposed algorithm.
>
> Ans: Thanks for the suggestions on a larger dataset for further evaluation. As suggested by reviewer 1, we intend to try other more challenging machine learning tasks such as speech recognition and computation vision in future works. This work focuses on the QTN-VQC based end-to-end learning framework for QNN, particularly its representation power which connects the number of qubits to the performance.
>
> ***
>
> **Q5.** When TT is used to approximate the fully connected weights, the gradient exploding and diminishing often occur due to many core tensors. But the paper has not presented any algorithm-related study and discussion.
>
> **Ans:** Thanks for the reviewer's comment on that issue. We admit that the gradient exploding and diminishing problems are serious issues in the TTN training procedures. Since we just consider three core tensors in our experiments and only one TT layer is involved, the related potential training problems are not the key concern in this work. In fact, we intend to use the most simple but a parametric TTN to extract quantum embedding for QNN.

---

> > ### Author Response · Authors · 2021-11-19
> > **Reply to Reviewer JaBi (2/2)**
> >
> > **Q6.** Input data x is reshaped into a k-order tensor, and then it is represented as TT format. However, this is the procedure of TT decomposition of data x, which is still a challenging task and computationally expensive.
> >
> > **Ans:** Thanks again for pointing that out. However, as we mentioned in the above response, a 3-order TTN with the TT-ranks (1, 2, 2, 1) is used in the experiments, which ensures the minimum computational cost for TTN such that the overhead incurred by the conversion from vectors to tensors. Table 6 is used to compare the absolute running time of QTN-VQC with AlexNet-VQC and Dense-VQC, and it shows that the computation speed of QTN-VQC lies in between Dense-VQC and AlexNet-VQC. Our results suggest that QTN-VQC is marginally slower than Dense-VQC, but it runs much faster than AlexNet-VQC.
> >
> > ***
> >
> > **Q7.** In Theorem 3, the upper bound is not related to TT-ranks, which seems not reasonable. Let us assume TTN with tt-ranks are all 1, can we achieve the same upper bound as a dense layer?
> >
> > **Ans:** Thanks for the reviewer’s comment on our theory. However, the upper bound is related to the order of tensors instead of the specified TT-ranks. The reason is that the matricization procedure can cancel the TT-ranks, so TT-ranks are not shown in the final upper bound.
> >
> > ***
> >
> > **Q8.** The paper is an improvement of the quantum neural network algorithm. The idea is simple but interesting for the machine learning community. However, novelty is fair, and the experimental evaluation is less convincing. Some important problems like computational cost, gradient exploding are not well discussed.
> >
> > **Ans:** Thanks for the summarization of our work. However, we do **not agree** with this related comments on our work.
> >
> > The main contribution of this work is to **leverage a TTN to compose a QTN-VQC model for the end-to-end learning paradigm of QNN. The use of TTN realizes a classical simulation of quantum machine learning algorithms**. Moreover, we delicately compose a TPE with quantum circuits to generate quantum embedding. Since we are only concerned with the most simple TTN with only one TT layer and TT-ranks as {1, 2, 2, 1}, the issue of gradient exploding is not the major concern in this work. Furthermore, we present some new experimental results on various TT-ranks and we also compare the convergence rates during the training process.

---

> > > ### Comment · Reviewer_JaBi · 2021-11-26
> > > **Reply to authors' response**
> > >
> > > Thanks for the authors's detailed responses and additional experiments.
> > >
> > > However, my main concern about the novelty and unconvincing evaluation are still unsolved.  The authors have leveraged a parametric TTN for quantum embedding generation such that an end-to-end learning framework can be built for quantum neural networks (QNN), however, this is the combination of several existing works, which seems straightforward.
> > >
> > > It is fine that the performance cannot beat the state-of-the-art performance by NNs, but the current performance on MNIST seems very low, which is even worse than traditional ML method like SVM.  Also TT decomposition for every data point is computational costly, especially for large number of training data.

---

### Official Review · Reviewer_sUwU · 2021-10-31

**Correctness:** 2
**Technical Novelty And Significance:** 2
**Empirical Novelty And Significance:** 1
**Recommendation:** 3
**Confidence:** 4

**Main Review:**

The QML community has already explored many different ways for encoding classical input data into the parameterized angles in the quantum embedding circuit. For example, in [1], they have considered using ResNet for doing the encoding. It is not clear from this work that using the tensor-train network is beneficial compared to the plethora of existing methods (such as ResNet). The improvement over using a dense neural network is also marginal (+3% to 4% on MNIST).

Furthermore, we can see that the prediction performance on MNIST is significantly worse than the state-of-the-art results (> 99% accuracy). A straightforward QML model is to consider a state-of-the-art CNN for MNIST and encode the output of the CNN into a single-qubit quantum circuit. We consider the single-qubit quantum circuit to act as an identity (so the quantum circuit is not doing anything). The performance of such a QML model will also achieve >99% accuracy because this trivial QML model is equivalent to the state-of-the-art CNN.

Together I think the main claim that "Our experiments on the MNIST dataset demonstrate the advantages of QTN for quantum embedding over other quantum embedding approaches." is unjustified.

[1] Lloyd, Seth, et al. "Quantum embeddings for machine learning." arXiv preprint arXiv:2001.036

**Summary Of The Paper:**

Quantum neural networks have two parts: a quantum embedding circuit that takes in classical data and embeds it into a quantum state, and a variational quantum circuit that learns a quantum circuit for evolving the quantum state before measurement.

The authors propose to encode the classical input data into the parameterized angle in the quantum embedding circuit using a tensor-train network (instead of a dense neural network). This results in a 3-4% improvement in prediction accuracy for MNIST (87.12% accuracy).

**Summary Of The Review:**

It is not clear that the proposed method is better than the existing QML models. Hence, the main claim of this work is not justified.

---

> ### Author Response · Authors · 2021-11-19
> **Reply to Reviewer sUwU**
>
> **Q1.** The QML community has already explored many different ways for encoding classical input data into the parameterized angles in the quantum embedding circuit. For example, in [1], they have considered using ResNet for doing the encoding. It is not clear from this work that using the tensor-train network is beneficial compared to the plethora of existing methods (such as ResNet). The improvement over using a dense neural network is also marginal (+3% to 4% on MNIST).
>
> **Ans:** Thanks for the suggestions on other approaches for feature extraction and dimension. To highlight the advantages of QTN-VQC, we also attempt the standard AlexNet for dimension reduction (though it still has about 10^4 times more parameters than QTN).
>
> Our simulation results in Table 1 shows that although AlexNet-VQC can achieve better performance than QTN-VQC in the condition of noiseless quantum circuits (91.43% vs. 92.81%), our results in Table 2 suggest that the QTN-VQC **outperforms the AlexNet-VQC** in the noisy condition of quantum circuits (89.32% vs. 87.86%). Moreover, we investigate the performance by applying different activation functions for TTN and utilizing more qubits for VQC (higher feature dimension from TTN). The simulation results in Table 3 demonstrate that the Sigmoid function can contribute to the best performance in our experiments.
>
> ***
>
> **Q2.** Furthermore, we can see that the prediction performance on MNIST is significantly worse than the state-of-the-art results (>99% accuracy). A straightforward QML model is to consider a state-of-the-art CNN for MNIST and encode the output of the CNN into a single-qubit quantum circuit. We consider the single-qubit quantum circuit to act as an identity (so the quantum circuit is not doing anything). The performance of such a QML model will also achieve >99% accuracy because this trivial QML model is equivalent to the state-of-the-art CNN.
>
> **Ans:** Thanks again for the comments. We further improve the baseline results of QTN-VQC by applying the Sigmoid activation function instead of Tanh, but the current baseline results are still lower than the state-of-the-art performance.
>
> As discussed in the tutorial for TensorFlow quantum (https://www.tensorflow.org/quantum/tutorials/mnist), where similar empirical results are obtained on the MNIST dataset, and quantum neural network is difficult to beat a classical neural network for classical data.
>
> ***
>
> **Q3.** Together I think the main claim is that "Our experiments on the MNIST dataset demonstrate the advantages of QTN for quantum embedding over other quantum embedding approaches." is unjustified.
>
> **Ans:** We strengthen our claims by taking into account more models and different experimental settings. In more detail, we empirically compare the performance of QTN-VQC with AlexNet-VQC in Tables 1 and 2, and we also consider the effects of more activation functions as shown in Table 3 and more number of qubits in Table 4.
>
> The discussion of computational cost and training convergence rates are also concerned in Tables 5 and 6 in Appendix B. Besides, we attempt a new experiment based on the LFW dataset to assess our QTN-VQC model, which is shown in Tables 7 and 8. In doing so, our experiments can better corroborate our theoretical analysis.

---

> > ### Comment · Reviewer_sUwU · 2021-11-19
> > **Reply to author comments**
> >
> > I appreciate the author's effort in improving the manuscript. Also, I absolutely agree with the nearly-impossible challenge in using quantum neural networks to achieve state-of-the-art in classical datasets. Currently, there is simply no evidence (even theoretical ones) that quantum neural networks are useful for commonly used classical datasets (in NLP and computer vision). To be honest, the hope of using near-term QNN to achieve an advantage in NLP or computer vision is even more elusive than finding axions, where we at least have a good theoretical understanding of.
> >
> > The current hope is that quantum ML may be beneficial for classical datasets that have a quantum origin (physics, chemistry, material science, etc). Even this is not justified in the near-term due to the power of classical techniques (both ML and non-ML).
> >
> > Because the theoretical progress and the practical implication of this work are both very limited, I retain my original score.

---

### Official Review · Reviewer_nHEv · 2021-11-04

**Correctness:** 3
**Technical Novelty And Significance:** 2
**Empirical Novelty And Significance:** 2
**Recommendation:** 5
**Confidence:** 4

**Main Review:**

- The experiments show an improvement in performance from using TTN for dimensionality reduction, but these results would be stronger if another dataset besides MNIST was used, and if less trivial baselines were used.

- There is very little new theoretical contributions. The tensor product encoding presented in theorem 1 has been extensively used before (since \[Stoudenmire and Schwab 2016\] at least), and theorems 2 and 3 are simple consequences of the universal approximation properties of dense shallow neural networks and tensor trains.

- The use of TTN as a dimensionality reduction method is poorly justified in the paper, and not compared against sufficient baselines in the experiments. In particular, given that the dimensionality reduction is intended to be fully classical, it would make sense to use less trivial classical models as experimental baselines (e.g. a neural net with more than one layer).

- The parameter counts given in Tables 1 and 2 are somewhat deceptive, as the stated requirement that input data is first converted into a TT format will involve a significant memory and runtime overhead. Converting data to TT format also seems unnecessary, and differs from the standard procedure for using matrices in TT format in place of dense matrices (e.g. \[Novikov et al. 2015\]).

**Summary Of The Paper:**

A two-stage framework for quantum machine learning is presented, which uses a tensor train network (TTN) to reduce the dimensionality of input data, whose embedded form is fed to a variational quantum circuit (VQC) and measured to obtain an output. Several theoretical results about this framework are presented, and experimental results show the improved performance of this framework against a pair of similar baseline models.

**Summary Of The Review:**

The paper's proposed method is a straightforward combination of several well-studied ideas, namely tensorization of linear weights, tensor product embeddings, and variational quantum circuits. The biggest strength of the paper is in its experimental results, but these consist of a few experiments on MNIST with a limited number of baseline models.

---

> ### Author Response · Authors · 2021-11-19
> **Reply to Reviewer nHEv (1/2)**
>
> **Q1** There are very few new theoretical contributions. The tensor product encoding presented in theorem 1 has been extensively used before (since [Stoudenmire and Schwab 2016] at least), and theorems 2 and 3 are simple consequences of the universal approximation properties of dense shallow neural networks and tensor trains.
>
> **Ans:** Thank you for the comments. However, we respectfully **disagree** that our work has few new theoretical contributions. For one thing, our main contribution in this work is to leverage the tensor-train network (TTN) to constitute a trainable quantum embedding method such that a **genuine end-to-end learning paradigm can be realized for quantum neural networks**.
>
> For another, although the mathematical expression of quantum encoding in [Stoudenmire, 2016] is matched with our tensor product encoding, we delicately design quantum circuits to realize the quantum encoding which is not shown in [Stoudenmire, 2016]. In more detail, although Eq. (6) has been shown in [Stoudenmire, 2016], how to leverage quantum circuits to realize Eq. (6) is still unknown, and we intend to implement the quantum encoding plan by employing Pauli-Y gates in our work. More importantly, we **analyze the presentation power of QTN-VQC by leveraging the Lemma of universal approximation theory, which has never been shown in other quantum machine learning works.**
>
> ***
>
> **Q2.** The use of TTN as a dimensionality reduction method is poorly justified in the paper, and not compared against sufficient baselines in the experiments. In particular, given that the dimensionality reduction is intended to be fully classical, it would make sense to use less trivial classical models as experimental baselines (e.g. a neural net with more than one layer).
>
> **Ans:** Thank you for the suggestions on the experimental parts. To highlight the advantages of our proposed QTN-VQC, we also investigate the use of AlexNet for dimension reduction. Note that the comparison between QTN-VQC and AlexNet-VQC is not fair to our proposal because AlexNet-VQC has 10^4 times more parameters than QTN-VQC.
>
> In our simulation in both noiseless and noisy conditions, although Table 1 shows that AlexNet-VQC exhibits better performance than QTN-VQC in the setting of noiseless quantum circuits, Table 2 presents that QTN-VQC outperforms AlexNet-VQC if the noisy quantum circuits are considered. Besides, the number of QTN-VQC parameters is much fewer than AlextNet-VQC.
>
> ***
>
> **Q3.** The parameter counts given in Tables 1 and 2 are somewhat deceptive, as the stated requirement that input data is first converted into a TT format will involve a significant memory and runtime overhead. Converting data to TT format also seems unnecessary, and differs from the standard procedure for using matrices in TT format in place of dense matrices (e.g. [Novikov et al. 2015]).
>
> **Ans:** Thanks for the comments. However, we do not agree that Tables 1 and 2 are misleading. For one thing, the memory overhead of TTN is much less than the related dense layer because TTN only has $\sum\limits_{k}r_{k}r_{k-1}I_{k}J_{k}$ parameters, which is a contrast to $\prod\limits_{k} I_{k}J_{k}$ parameters for a dense layer. For another, as shown in our new analysis in Appendix B.2, the computational cost of TTN can become minimal if small TT-ranks and order K can be set. In our experiments, we set K as 3 and configure TT-ranks as {1, 2, 2, 1}, which corresponds to the minimal computational cost for TTN.
>
> Besides, as shown in our analysis in Eqs. (3) and (4), it is necessary to decompose the input vector into the TT format, which is supported by the 2nd and 3rd lines in Eq. (4) and strictly consistent with [Novikov et al., 2015].

---

> > ### Author Response · Authors · 2021-11-19
> > **Reply to Reviewer nHEv (2/2)**
> >
> > **Q4.** The experiments show an improvement in performance from using TTN for dimensionality reduction, but these results would be stronger if another dataset besides MNIST was used, and if less trivial baselines were used.
> >
> > **Ans:** Thank you so much for the suggestions.
> >
> > First, we need to highlight that unlike classical machine learning, the evaluation of the full MNIST dataset is a challenging task for quantum machine learning algorithms. Many state-of-the-art works such as [Wang et al., 2021] and [Chen et al, 2021] employ 2 digits recognition in the MNIST dataset, where 92%-99% accuracy can be attained. However, it is difficult to maintain high accuracy when 10 digits are concerned. An existing result on the full MNIST dataset is shown on the website (https://www.tensorflow.org/quantum/tutorials/mnist), which is related to the example of TensorFlow quantum and around 90% accuracy can be obtained for the full MNIST dataset.
> > Besides, in this work, we concentrate on designing quantum tensor networks for an end-to-end learning framework for quantum neural networks.
> >
> > ***
> >
> > **References:**
> >
> > [1] Wang, H., Ding, Y., Gu, J., Lin, Y., Pan, D. Z., Chong, F. T., & Han, S. (2021). QuantumNAS: Noise-adaptive search for robust quantum circuits. arXiv preprint arXiv:2107.10845.
> >
> > [2] Chen, S. Y. C., Huang, C. M., Hsing, C. W., & Kao, Y. J. (2021). An end-to-end trainable hybrid classical-quantum classifier. arXiv preprint arXiv:2102.02416.

---

> > > ### Comment · Reviewer_nHEv · 2021-11-23
> > > **Thank you for your detailed response**
> > >
> > > I want to thank the authors for the time put into their detailed response to my concerns on the manuscript, and have raised my score to account for the new additions to the manuscript. I have a few follow-up comments in reply to the authors response:
> > >
> > > **Answer 2 (A2):** To highlight the advantages of our proposed QTN-VQC, we also investigate the use of AlexNet for dimension reduction. Note that the comparison between QTN-VQC and AlexNet-VQC is not fair to our proposal because AlexNet-VQC has 10^4 times more parameters than QTN-VQC.
> > >
> > > **Response 2 (R2):** Thank you for adding this baseline! I agree that this comparison isn't fair to your QTN-VQC architecture, and it might be more useful to the reader to provide a somewhat *less* non-trivial baseline here, in order to permit a more equal comparison (e.g. a CNN with fewer layers of smaller dimension than AlexNet). Although there might not be pretrained examples of such networks easily available, you could always train one jointly with your other network parameters.
> > >
> > > **A3:** Thanks for the comments. However, we do not agree that Tables 1 and 2 are misleading. For one thing, the memory overhead of TTN is much less than the related dense layer... Besides, as shown in our analysis in Eqs. (3) and (4), it is necessary to decompose the input vector into the TT format, which is supported by the 2nd and 3rd lines in Eq. (4) and strictly consistent with [Novikov et al., 2015].
> > >
> > > **R3:** To be clear, I absolutely agree with your point about the parameter count of TTN being smaller than an equivalent dense matrix, and this serves as one of the motivating advantages of using TTNs as a means of "tensorizing" neural network weights. The point I was trying to make is that your paper (and response) states that you must first convert each input vector from a dense format into a TT format, and this operation (typically carried out using the TT-SVD algorithm) requires resources that scale super-linearly with the dimension of the input vector. While you claim that this requirement is consistent with [Novikov et al., 2015], they in fact **don't** carry out this conversion, and instead use dense input vectors without any conversion to TT format (note the differences between Equation 5 of their paper to Equation 4 of yours). This allows for a faster TT evaluation overall, and I would encourage you to follow their lead in using the dense representation of inputs in QTN-VQC.

---

> > > > ### Author Response · Authors · 2021-11-24
> > > > **Further reply to Reviewer nHEv**
> > > >
> > > > **Ans**: Thank you again for the new comments. For sure, we also set up a new model of CNN-VQC to compare with our proposed model QTN-VQC. The architecture of CNN-VQC follows the setup as: conv_layer(10 chans) — conv_layer(20 chans) — FC(320 x 8), where the kernel_size is 5. The experimental results in Table I and II are modified as follows:
> > > > ```
> > > > Table I: Empirical results on the MNIST test dataset under the noiseless quantum circuit setting
> > > > Models		          Params	           CE		 	   Acc(%)
> > > > PCA-VQC                 144                0.5877            82.48 \pm 1.02
> > > > Dense-VQC               6416               0.4132            88.54 \pm 0.73
> > > > AlexNet-VQC             3.25x10^{6}        0.2562            92.81 \pm 0.47
> > > > CNN-VQC                 7992               0.3578            90.12 \pm 0.49
> > > > QTN-VQC                 328                0.3090            91.43 \pm 0.51
> > > > ```
> > > > ```
> > > > Models		Params		  Acc_{q20}(%)		Acc_{depo}(%)
> > > > PCA-VQC		144		    81.23 \pm 1.34		 81.98 \pm 1.17
> > > > Dense-VQC	  6416		   84.55 \pm 1.22		 86.09 \pm 1.04
> > > > AlexNet-VQC	3.25x10^{6}	87.46 \pm 1.34		 87.86 \pm 1.08
> > > > CNN-VQC	    7992		   87.21 \pm 1.26		 87.45 \pm 1.02
> > > > QTN-VQC		328			88.12 \pm 1.09		 89.32 \pm 1.07
> > > > ```
> > > >
> > > >
> > > > **R3**: To be clear, I absolutely agree with your point about the parameter count of TTN being smaller than an equivalent dense matrix, and this serves as one of the motivating advantages of using TTNs as a means of "tensorizing" neural network weights. The point I was trying to make is that your paper (and response) states that you must first convert each input vector from a dense format into a TT format, and this operation (typically carried out using the TT-SVD algorithm) requires resources that scale super-linearly with the dimension of the input vector. While you claim that this requirement is consistent with [Novikov et al., 2015], they in fact don't carry out this conversion, and instead use dense input vectors without any conversion to TT format (note the differences between Equation 5 of their paper to Equation 4 of yours). This allows for a faster TT evaluation overall, and I would encourage you to follow their lead in using the dense representation of inputs in QTN-VQC.
> > > >
> > > > **Ans**: Thank you for pointing that out. We changed the related implementation of TT computation to speed up the running time.

---

### Author Response · Authors · 2021-11-19
**Official Summary of Updates**

We would like to sincerely thank all reviewers for the high-quality reviews and constructive feedback. We have revised our draft according to the comments. Revisions are highlighted in blue in the new version.

Below we provide a summary of updates:

***

**1.** We highlight the main contribution of this work is to leverage Tensor-Train Network (TTN) to realize an end-to-end learning framework for quantum neural networks.

The TTN does not simply aim at a replacement of a fully-connected layer, but it provides a tractable tensor network to classically simulate the quantum machine learning algorithms because tensor networks can find the replacement of quantum circuits which can approximate quantum state vectors.

**2.** We highlight our contribution more clearly to the tensor product encoding. In more detail, we bridge the gap between an explicit implementation of quantum circuits and the mathematical equation that has been shown before.

As suggested by the reviewers, we attempt an AlexNet to set up an advanced deep learning structure for dimension reduction, and accordingly, we build an AlexNet-VQC to compare the performance with QTN-VQC.

**3.** We further discuss the computational cost of QTN-VQC and the convergence rates of the training process, which are attached to Appendix B of the updated paper.

**4.** We highlight the challenge of a full MNIST dataset for quantum machine learning algorithms.
New results of different activation functions are shown in Table 3.

**5.** We simulate the performance with more qubits for the QNN models, the results are presented in Table 4.
A new experiment based on the LFW dataset is attempting to corroborate our theoretical analysis in Appendix C.

---

### Decision · Program_Chairs · 2022-01-20

**Decision:**

Reject

**Comment:**

This submission proposes a trainable quantum tensor network (QTN) for quantum embedding generation on a variational quantum circuit (VQC), which is followed by a significant empirical study on the QTN-VQC performance on the MNIST test dataset. However, the discussion during the review period raises some serious concerns about both the theoretical and empirical contributions of the submission.

On the theory side, we acknowledge the authors propose the use of tensor-train-network (TTN) as the dimension reduction layer of input features, design the tensor-product-encoding (TPE), and characterize the representation power of QTN-VQC.  However, a general feeling among all reviewers is that these contributions are kind of observational, rather than deep theoretical insights, based on existing results. For example, TTN is a well-established tool in dimension reduction, and the representation power of QTN-VQC can be derived as simple corollaries of the existing universal approximation theorem for a feed-forward neural network.  Moreover, the authors emphasize a lot that the use of TTN allows a genuine end-to-end quantum implementation of QTN-VQC because TTN can be relatively easier to implement as quantum circuits. However, the benefit of this end-to-end quantum implementation is rather unclear, especially for NISQ applications. In particular, because NISQ machines have limited quantum resources, should not a classical-quantum hybrid implementation be preferred in the interest of saving quantum resources? As genuine quantum implementation becomes the major motivation for using TTN, there is no other theoretical justification for selecting TTN.

We appreciate the authors’ efforts in carrying out the empirical study and the active interaction during the discussion period. Unfortunately,  most of the reviewers are not convinced by the existing experimental results. This is partially because the authors seek to support a very strong claim that QTN-VQC would outperform the state-of-the-art classical solutions, whereas the limitation of current quantum devices prevents any empirical study of QTN-VQC at the scale comparable to the classical solutions. Moreover, as pointed out by one of the reviewers, there is hardly any existing evidence that quantum neural networks would be useful at all for commonly used datasets in NLP and vision. Given that, it is very unlikely that a conclusion could be reached by an empirical study on small-scale instances. Nevertheless, since the authors aim to compare QTN with other quantum embedding methods,  other kinds of experiments are possible without directly comparing with the state-of-the-art classical solutions.

We believe that the submission would benefit a lot from addressing the concerns for both the theory and empirical parts and hope the authors would pursue it.